# Geography as a Determinant of Health: Health Services Utilization of Pediatric Respiratory Illness in a Canadian Province

**DOI:** 10.3390/ijerph18168347

**Published:** 2021-08-06

**Authors:** Shehzad Kassam, Jesus Serrano-Lomelin, Anne Hicks, Susan Crawford, Jeffrey A. Bakal, Maria B. Ospina

**Affiliations:** 1Department of Family Medicine, Faculty of Medicine & Dentistry, University of Alberta, Edmonton, AB T6G 2R3, Canada; shehzada@ualberta.ca; 2Department of Obstetrics & Gynecology, Faculty of Medicine & Dentistry, University of Alberta, Edmonton, AB T6G 2S2, Canada; jaserran@ualberta.ca; 3Department of Pediatrics, Faculty of Medicine & Dentistry, University of Alberta, Edmonton, AB T6G 1C9, Canada; eagreene@ualberta.ca; 4Alberta Perinatal Health Program, Alberta Health Services, Edmonton, AB T2N 2T9, Canada; Susan.Crawford@albertahealthservices.ca; 5Provincial Research Data Services, Alberta Health Services, Edmonton, AB T6G 2C8, Canada; jbakal@ualberta.ca

**Keywords:** respiratory diseases, pediatrics, geography, health inequalities

## Abstract

Respiratory diseases contribute to high healthcare utilization rates among children. Although social inequalities play a major role in these conditions, little is known about the impact of geography as a determinant of health, particularly with regard to the difference between rural and urban centers. A regional geographic analysis was conducted using health repository data on singleton births between 2005 and 2010 in Alberta, Canada. Data were aggregated according to regional health sub-zones in the province and standardized prevalence ratios (SPRs) were determined for eight respiratory diseases (asthma, influenza, bronchitis, bronchiolitis, croup, pneumonia, and other upper and other lower respiratory tract infections). The results indicate that there are higher rates of healthcare utilization in northern compared to southern regions and in rural and remote regions compared to urban ones, after accounting for both material and social deprivation. Geography plays a role in discrepancies of healthcare utilization for pediatric respiratory diseases, and this can be used to inform the provision of health services and resource allocation across various regions.

## 1. Introduction

Respiratory illnesses are one of the leading causes of emergency department (ED) visits and hospitalizations among children under the age of five in Canada [1]. This results in a significant burden to individuals and families, as well as the healthcare system. These illnesses include infectious diseases, such as pneumonia, influenza, and bronchiolitis, as well as inflammatory diseases, such as asthma.

The etiology and risk factors associated with pediatric respiratory illnesses are multifaceted, and while the biomedical model is well-established in terms of understanding these conditions, exploration of the social determinants of health (SDOHs) is more recent. SDOHs are the social, economic, and cultural factors that impact health at the individual and population levels [2]. Several studies have highlighted that low socio-economic status (SES) and low education attainment are strongly associated with asthma and other respiratory diseases [3,4,5]. In addition, housing conditions are a considerable determinant for these diseases because of overcrowding, the need for major repairs, and compromised indoor air quality [6,7,8,9,10,11].

The place where people live, as a determinant of health, has not been thoroughly researched in terms of pediatric respiratory illnesses in Canada. Health geographic analyses explore the spatial heterogeneity of diseases and provide information not only about disease distribution, but about also the factors and mechanisms associated with the risk of infection or development of a disease [12,13,14]. Given the extensive land area of Canada, evaluating geographic variability and its association with health can offer insight into unique challenges that affect communities most impacted by this determinant, specifically those living in rural and remote regions [12].

Rural communities face significant barriers in terms of lack of access to health services; patients often require substantial transportation to receive appropriate care. As a result, many individuals may delay seeking support until their condition deteriorates and, ultimately, require hospitalization [15,16]. With regard to respiratory diseases specifically, children in rural communities are exposed to more environmental factors that can precipitate or worsen conditions, creating disparities across the urban–rural gradient [17]. These include higher rates of smoking in rural households and occupational exposures to pesticides, dust, livestock, diesel fumes, and solvents in farming communities [5,18,19,20]. Some regions, particularly in the province of Alberta, also experience higher rates of industrial emissions from coal-fired power plant and petrochemical industry processes [21].

Recent studies have explored the intersections of geographic inequalities and pediatric respiratory illnesses in urban centers, noting discrepancies between various neighborhoods; however, little research has evaluated these factors across larger geographic regions, accounting for rural and remote communities [22]. The objective of this study was to explore the geographic inequalities in respiratory healthcare utilization during early childhood and potential links with economically or socially deprived populations in Alberta.

## 2. Materials and Methods

### 2.1. Study Design and Setting

This was a cross-sectional, secondary analysis using data from a retrospective birth cohort study of all single live births (≥22 weeks of gestation) that occurred in Alberta between 2005 and 2010, with follow-up until five years of age. Alberta is a province located in Western Canada, with a population of ~4 million people and a publicly funded, single-payer healthcare system [23]. The University of Alberta’s Health Research Ethics Board (Pro00088569) granted ethics approval for this research. The study is reported following recommendations by the Strengthening the Reporting of Observational Studies in Epidemiology (STROBE) statement [24].

### 2.2. Study Population and Data Sources

The original birth cohort included 206,994 singleton live births occurring in Alberta between 1 April 2005 and 31 March 2010 identified from the Alberta Perinatal Health Program, a clinical registry of all Alberta births attended at hospitals or by registered midwives at homes.

The original study flow diagram, data sources, and demographic characteristics of the cohort have been described elsewhere [22,25,26]. Briefly, we obtained de-identified, individual-level data from administrative health databases (i.e., Discharge Abstracts Database and the National Ambulatory Care Reporting System) on all respiratory hospitalizations and ED visits occurring from birth until 5 years of age for every member of the cohort. These administrative health databases capture sociodemographic information and data on diagnosis and clinical procedures for every episode of acute care using the International Classification of Diseases, Tenth Revision, enhanced Canadian version (ICD-10-CA), diagnostic codes [27].

### 2.3. Definition of Geographic Areas

Individual data on respiratory hospitalizations and ED visits occurring between ages 0 to 5 were aggregated into 35 large geographic areas (health subzones) consisting of populations >40,000 each using the postal code of residence at birth. Health subzones were created by Alberta healthcare authorities for the reporting of demographic, socio-economic, and population health statistics (e.g., health status, services utilization, care complexity rates) (Figure A1a) [28]. These 35 subzones belong to five geographic regions in which health services delivery is organized across the province: South, Calgary, Central, Edmonton, and North (see Table A1 and Figure A1c). The 35 subzones were chosen to aggregate data from low-populated rural areas dispersed across Alberta. Provincial demographic data for 2006–2016 indicates that about 65% of the population live in the metropolitan areas of Calgary and Edmonton, which represent only 2.2% of the total territory [29,30]. The remaining 35% of the population is scattered across the province in areas separated by large extensions of natural landscape (i.e., boreal forest in the north, the Rocky Mountains in the west, and grasslands in the south) [31].

For each member of the birth cohort, we linked the six-character postal code of the maternal place of residence at delivery to the corresponding dissemination areas (DAs), which are the smallest geographic areas for which census data is reported [32]. Dissemination area boundaries were defined using the 2006 census geography framework and the DMTI Spatial Postal Code Suite (Figure A1b) [33,34]. Second, a vector overlay union function in QGIS software was used to associate the DA boundaries with the 35 geographic subzones [35]. One DA including 1478 births was not geographically located within any subzone. Therefore, a total of 205,516 births (99.3% of the original study cohort) providing linkable individual data with the health subzones were selected for further analysis. The shapefiles of subzones’ geographic boundaries were provided by the Alberta Health Services (AHS) and are publicly available at http://www.ahw.gov.ab.ca/IHDA_Retrieval/ihdaGeographic.do (accessed on 28 July 2020).

### 2.4. Aggregation of Episodes of Acute Respiratory Healthcare Utilization to Health Subzones

All episodes of acute respiratory healthcare utilization were geographically located within 4384 DAs, from which 492 (9.2%) overlapped between two or more subzones. The total numbers of births and respiratory events in the DA across overlapping subzones were divided and weighted proportionally by the population size of each health subzone.

### 2.5. Study Outcomes: Respiratory Events

For each infant in the birth cohort, we obtained data on all events of acute healthcare utilization between birth and five years of age with an ICD-10-CA primary diagnostic code indicative of any of the following respiratory conditions: acute bronchitis (J20), asthma (J45), bronchiolitis (J21), croup (J05), influenza (J09–J11), pneumonia (J12–J18), other acute lower respiratory tract infections (J22), and other acute upper respiratory tract infections (J00–J06, except J05). We merged recurrent wheezing (R06.2) events with asthma or bronchiolitis based on the most prevalent respiratory condition after the first wheezing event. The follow-up period for respiratory acute healthcare services ran from 2005 to 2015, with data censoring at death or the end of the follow-up period (i.e., five years of age).

### 2.6. Socioeconomic Status

Material and social deprivation indices derived from 2006 census data were used as area-level proxy measures of SES [36,37]. These indices are composite measures that integrate DA census data for the population in the province aged 15 and over, excluding First Nations groups. DA-level data on income, education, and employment compose the material deprivation index, whereas marital status, one-person household, and single-parent family information compose the social deprivation index. They are reported in quintiles (Q_1_ = least deprived to Q_5_ = most deprived). Census data for 2006 was preferred over 2011 census data for index calculations as the latter resulted in a high global nonresponse rate [38].

Subzone-level indicators of material and social deprivation were calculated as the proportion of Q_4_ and Q_5_ DAs (the two most deprived quintiles). These indices were used in the statistical analysis to correlate with acute respiratory healthcare utilization by subzone.

### 2.7. Statistical Analysis

Descriptive statistics were calculated for the number of births and the total number and rates of respiratory healthcare services aggregated by the five larger regions and the 35 subzones.

Geographic inequalities of respiratory healthcare utilization events were evaluated through the comparison of standardized prevalence ratios (SPRs) for single and aggregated (total) respiratory outcomes across the various subzones [39]. For each subzone, the SPR was calculated by dividing the total number of respiratory healthcare utilization events by the “expected” number of respiratory healthcare utilization events. The latter was calculated by multiplying the provincial rate of respiratory healthcare utilization events by the number of births. For the provincial rate, we used the total number of respiratory health services during the study period as the numerator and the total number of births as the denominator. An SPR greater than one indicates that more respiratory healthcare utilization events were observed than expected. The means and 95% confidence intervals (CIs) of SPRs, combining all respiratory outcomes by subzone, are graphically reported. The SPRs for all respiratory healthcare utilization events and for each respiratory illness for all 35 health subzones were mapped using choropleth (descriptive) maps. We used the Jens natural breaks classification as a clustering method to determine the best arrangement of values into different classes while minimizing the squared deviations of the class means and maximizing between-class differences [40]. Choropleth maps were also used to display the geographic distribution of both material and social deprivation indices by subzones.

Finally, correlations between the SPRs (all respiratory healthcare utilization events and for each respiratory illness by all 35 health subzones) and the material/social deprivation indices of subzones were described using the Spearman correlation index. We used this non-spatial correlation approach instead of alternative spatial methods because the estimation of spatial autocorrelation among contiguous areas surrounded by large extensions of unpopulated natural landscape (as previously explained) can be unrealistic. All statistical analyses were performed using STATA, release 15 [41]. Choropleth maps were created using QGIS [35].

## 3. Results

A total of 297,306 healthcare utilization events from 205,516 singleton live births (99.3% of the original study cohort), which provided linkable individual data with the health subzones (Figure 1), were included in the analysis. Data on the total number of births, total number of respiratory healthcare utilization events, and crude rates of events for each individual respiratory illness for all 35 health subzones are shown in Table A2. Choropleth maps of live births and the SPR of total events (aggregation of ED visits and hospitalizations for all respiratory illnesses) are shown in Figure 2. The choropleth map of live births (Figure 2A) indicates that the greatest number of singleton live births occurred in the large urban centers of Edmonton and Calgary, and the fewest occurred in southern and other rural subzones. The SPRs of healthcare utilization for the pediatric respiratory illnesses (Figure 2B) indicated that the greatest proportion of total healthcare utilization events occurred in the northern subzones (2.06–3.11), with the fewest occurring in the urban centers (0.46–0.73).

### 3.1. Respiratory Healthcare Utilization

#### 3.1.1. Regional (Zone)

Table 1 indicates the singleton live births and total healthcare utilization events for each zone, and the percentage of those as compared to the entire province, as well as the rate of events per birth. The lowest proportion of births was found for the South (4.79%), North (10.38%), and Central (10.59%) zones, while the greatest proportion occurred in the Calgary (41.25%) and Edmonton (32.98%) zones. The greatest proportion of healthcare utilization events occurred in the Calgary (31.73%), North (24.96%), and Edmonton (22.62%) zones, while the smallest proportion occurred in the South (5.81%) and Central (14.89%) zones. The provincial rate of events per birth was 1.45. The Edmonton (0.99) and Calgary (1.11) zones had lower rates than the entire province. The North (3.48), Central (2.03), and South (1.75) zones had greater rates than the entire province.

#### 3.1.2. Respiratory Illness

Respiratory conditions with the highest healthcare utilization included other acute upper respiratory tract infections (oURTI) (52.00%; *n* = 154,606), croup (11.57%; *n* = 34,389), asthma (9.67%; *n* = 28,757), pneumonia (9.26%; *n* = 27,533), and bronchiolitis (8.37%; *n* = 24,895). The conditions with the lowest healthcare utilization included other acute lower respiratory tract infections (oLRTI) (1.72%; *n* = 5119), influenza (2.49%; *n* = 7424), and bronchitis (4.87%; *n* = 14,488).

#### 3.1.3. Standardized Prevalence Ratio

The rate of respiratory healthcare events in Alberta was 1.45 per singleton live birth. The range of SPR values for all subzones and respiratory illnesses was 0.14 (bronchitis, Z2.1 and Z2.2) to 4.80 (bronchitis, Z5.3). Table 2 indicates the SPRs by subzone for healthcare utilization events for all respiratory illnesses combined and for each individual respiratory illness. Figure 3 shows the mean SPRs with 95% CIs by subzone for all healthcare utilization events for respiratory illnesses. In both Table 2 and Figure 3, the reference SPR for the entire province was 1.0. As indicated in the table and figure, there were subzones in the Central zone where the SPR was roughly two times greater and subzones in the North zone where the SPR was roughly three times greater than the provincial SPR. In addition, subzones in both the Calgary and Edmonton zones fell in line with or were less than the provincial SPR. Overall, greater rates of ED visits and hospitalizations for children aged 0–5 were noted in northern communities compared to southern ones, as well as in rural regions compared to urban ones. The choropleth maps of the SPRs by subzone for healthcare utilization events for individual respiratory illnesses are shown in Figure A2.

### 3.2. Material and Social Deprivation

Correlations between the SPRs (for healthcare utilization events for all respiratory illnesses combined and for each individual respiratory illness) and the material and social deprivation indices (utilizing quintiles Q_4_ and Q_5_, the most deprived levels for all subzones) are shown in Table 3. There was a moderate positive correlation (0.48) between the total number of healthcare utilization events and material deprivation, with a range from 0.36 to 0.51. There was also a weak negative correlation (−0.29) between these events and social deprivation, with a range from −0.05 to −0.36. [42]. Choropleth maps of the material and social deprivation indices by subzone are shown in Figure A3.

## 4. Discussion

This study analyzed the regional distribution of respiratory healthcare utilization in early childhood in Alberta and the correlation with indicators of material and social deprivation. Geographic inequalities are evident in the distribution of healthcare utilization for pediatric respiratory illness across the province, with greater rates occurring in northern communities, as well as rural and remote regions. While there is some association with material and social deprivation for all respiratory illnesses with respect to the healthcare utilization, other factors play a role in the discrepancies between subzones.

Three overarching themes, supported by current literature, may explain the trends of higher rates in northern and rural regions. These themes are community demographics, environmental risk factors, and access to preventive and primary healthcare services.

Material deprivation, which was found to have a moderate positive correlation with these conditions, may have a significant impact in these communities. An Alberta Population Health Profile from 2010 indicates that lower education levels, as well as lower median and average income, were present in northern communities compared to the rest of Alberta [43]. In addition, Indigenous communities are disproportionately affected by pediatric respiratory illnesses and a greater percentage of the population in northern communities identified as Indigenous (15.7%) compared to Alberta as a whole (5.8%) [43,44,45,46,47].

A multitude of environmental risk factors may contribute to the development and/or exacerbation of pediatric respiratory illnesses. Household and parental smoking are associated with increased rates and severity of disease for both upper and lower respiratory tract infections, including pneumonia, bronchiolitis, croup, and influenza [48,49]. The prevalence of smoking is higher in rural and northern regions in Alberta and across Canada. This may be due to fewer smoking restrictions in these communities, greater proportions of individuals working in manual labour occupations, and lower SES [48,50]. These factors result in greater secondary smoke exposure in children that live in these communities and may play a role in their respiratory conditions. Beyond household smoking, specific regions in Alberta are known to have greater industrial air pollutants [51]. Oil sands, several of which are situated in northern Alberta, emit higher concentrations of sulfur, nitrogen oxides, and particulate matter. These substances are known respiratory irritants, which may be contributors to both development and exacerbation of pediatric conditions [52,53,54].

Given the extensive range of latitudes in Alberta, climate differences exist across the province. While several regions have a humid continental climate in southern parts of Alberta, the vast majority of northern Alberta has subarctic or boreal climates [55]. A study in the province of Ontario highlighted that regions with colder climates have increased rates of respiratory illnesses, which may explain higher rates of these conditions in northern Alberta as well [56,57].

Health inequalities are common in rural and northern regions of Alberta, particularly in terms of adequate access to preventive and primary healthcare [50,58]. Rural communities with greater distances to large urban centers often have lower vaccination rates than their metropolitan counterparts [59,60]. In 2018–2019, the North zone had the lowest percentage of children vaccinated for influenza under the age of six (20%) and had the greatest rate of laboratory-confirmed influenza cases (295 per 100,000) in Alberta [61,62]. Given that both influenza and several upper and lower respiratory tract infections in children are preventable through vaccinations, this may result in higher rates in ED visits and hospitalizations in these regions. Several studies have also shown that rural and northern communities face barriers to accessing primary care [58,63,64]. They often require ED visits as their only means of obtaining healthcare services or may require these services due to the progressive severity of their conditions [65]. In 2017, AHS conducted surveillance of healthcare service visit rates across the province and showed that three distinct regions (Remote North, Rural North, and Remote West) had the highest rates of ED visits and hospital admissions in the province. These sites were also three of the four lowest regions for family doctor visits per year. In contrast, the largest urban centers in the province (Greater Edmonton and Greater Calgary areas) had the fewest ED visits and hospital admissions and the most family doctor visits per year [66]. Although this study focuses on pediatric respiratory diseases, there is a clear rural–urban gradient for healthcare utilization [67].

Several limitations exist within this study related to diagnostic codes, geographic boundaries, determinants of health, and inclusion of Indigenous communities. First, the use of ICD-10-CA codes may be limited by the clinician’s ability to label each presentation with the appropriate respiratory disease based on individual clinical acumen and/or access to diagnostic testing. The inclusion of more general codes for conditions such as “oURTI” or “oLRTI” can account for any diagnostic uncertainty; however, this may also pose as barrier for further community-level targeted interventions based on each condition’s specific risk factors. Second, large geographic regions (subzones) limit the ability to identify key high-rate regions as data are distributed over a greater population. Although some inaccuracies in the geographic analysis may have arisen, as 9.2% of dissemination areas overlapped between two or more subzones, this proportion is small when considering the entire range of data and unlikely to influence the study results. Third, although material and social deprivation indices were essential in our analysis, the provincial health databases utilized for this study were limited with respect to the broad range of determinants of health that are associated with pediatric respiratory illnesses, including ecological and cultural factors. Finally, given that the material and social deprivation indices were derived from national census data, the incomplete enumeration of many First Nations reserves and Metis settlements resulted in their lack of inclusion in these composite measures [68]. With Indigenous communities being heavily impacted by pediatric respiratory illnesses, as well as facing many barriers to overcome material and social inequities, these indices may underestimate the association with these pediatric respiratory health conditions in this study.

This study highlights geography as a determinant of health that is often underemphasized in research involving pediatric respiratory illnesses and healthcare utilization, with a particular focus on rural and remote communities. Given the extensive land area of Alberta, the use of geographic information systems, in association with regional health authority boundaries, allowed for unique means of understanding a geographic dimension of respiratory health inequalities affecting children across the province. The implications of evaluating distinct regions in terms of their risk factors of pediatric respiratory illnesses can be used to coordinate and apply future interventions in each respective zone. In addition, recognition of inequalities in healthcare utilization can guide resource allocation within the realms of preventive, primary, secondary, and tertiary healthcare. Strategies that can be employed to address these issues must occur at a population health level. Limited access to healthcare should be met with provisions to increase primary care providers in these communities, as well as utilization of telehealth services. In addition, health promotion campaigns would highlight the importance of preventive measures, such as immunizations, for individuals and families. Lastly, advocacy and policy development at the provincial level can support the reduction of environmental hazard emissions that heavily impact pediatric respiratory health. These approaches would not only address gaps across large geographic areas but also support underserviced populations who are disproportionately affected by these conditions (e.g., those with higher material deprivation, Indigenous and northern communities, etc.).

## 5. Conclusions

The purpose of this study was to explore geographic inequalities in healthcare utilization for pediatric respiratory illness in Alberta, Canada. Rural communities in Alberta, particularly in northern regions, face geographic inequalities related to these diseases as determined by ED and hospital-based healthcare utilization. Although material and social deprivation play a role, a multitude of factors, such as limited access to preventive and primary healthcare, greater rates of exposure to smoking and industrial air pollutants, colder climates, and specific barriers affecting Indigenous communities, exist as well. This study is a steppingstone for population-level translational research targeted at pediatric respiratory illnesses in rural and remote communities. Addressing issues of limited access to care, as well as mitigating risks associated with environmental health hazards from pollutants and climate, may reduce the burden of disease in these regions, particularly those of underserved communities. Future studies are warranted for further expansion of these findings.

## Figures and Tables

**Figure 1 ijerph-18-08347-f001:**
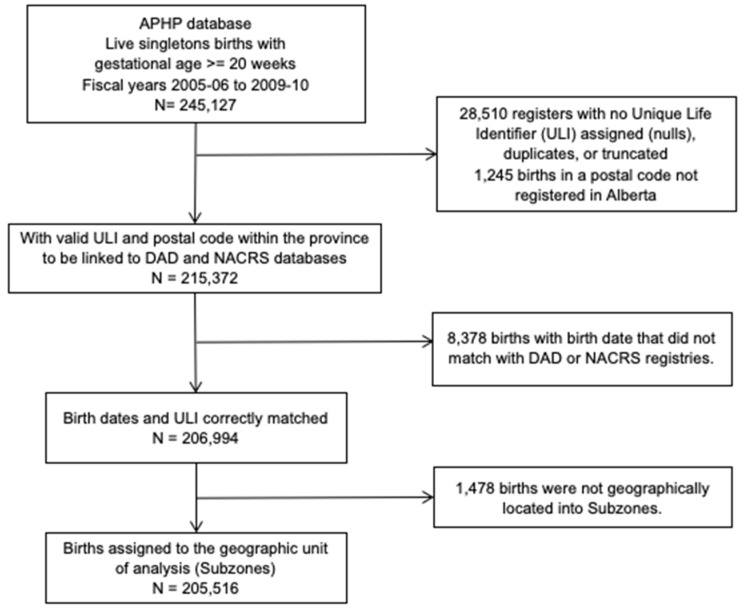
Flow diagram of study population of singleton live births in Alberta.

**Figure 2 ijerph-18-08347-f002:**
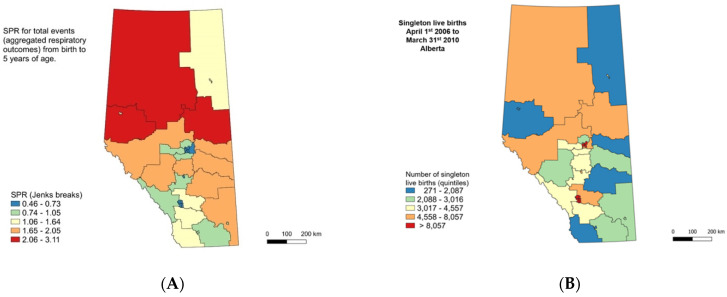
Choropleth maps of numbers of singleton live births organized into quintiles (**A**) and the SPRs of total healthcare utilization events for all respiratory illnesses combined and organized into quintiles using Jenks breaks (**B**) in Alberta.

**Figure 3 ijerph-18-08347-f003:**
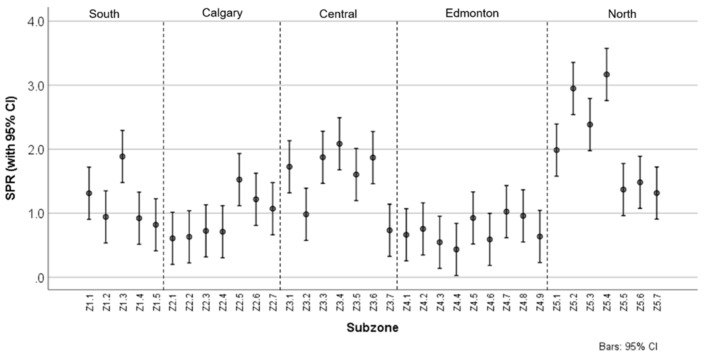
Means and 95% CIs of SPRs of all healthcare utilization events for pediatric respiratory illnesses by subzones in Alberta.

**Table 1 ijerph-18-08347-t001:** Singleton live births, total healthcare utilization events for all pediatric respiratory illnesses, and events per birth in Alberta and in each zone.

	Alberta	South Z1.1–1.5	Calgary Z2.1–2.7	Central Z3.1–3.7	Edmonton Z4.1–4.9	North Z5.1–5.7
Births, *n* (%)	205,516 (100%)	9841 (4.79%)	84,776 (41.25%)	21,758 (10.59%)	67,786 (32.98%)	21,335 (10.38%)
Events, *n* (%)	297,306 (100%)	17,263 (5.81%)	94,340 (31.73%)	44,258 (14.89%)	67,236 (22.62%)	74,209 (24.96%)
Events per birth	1.45	1.75	1.11	2.03	0.99	3.48

**Table 2 ijerph-18-08347-t002:** Heat map of standardized prevalence ratios of healthcare utilization events for pediatric respiratory illnesses by subzones in Alberta (heat map) ^1^.

Zone	Subzone	Total	Asthma	Bronchitis	Bronchiolitis	Croup	Influenza	oLRTI	oURTI	Pneumonia
**South**	Z1.1	1.54	1.36	1.04	1.11	1.24	1.81	0.91	1.81	1.22
	Z1.2	0.98	0.72	0.85	1.10	1.15	1.11	0.66	0.98	0.96
	Z1.3	2.00	1.21	3.17	1.39	1.81	2.45	1.56	2.31	1.19
	Z1.4	0.86	0.78	0.70	0.89	1.02	1.13	1.21	0.83	0.82
	Z1.5	0.79	0.99	0.51	1.00	0.91	0.81	0.41	0.66	1.28
**Calgary**	Z2.1	0.59	0.98	0.14	0.64	0.91	0.68	0.44	0.48	0.59
	Z2.2	0.62	1.07	0.14	0.65	0.60	0.80	0.54	0.55	0.70
	Z2.3	0.71	1.09	0.28	0.73	0.94	0.82	0.57	0.62	0.75
	Z2.4	0.71	1.00	0.28	0.69	0.96	0.81	0.53	0.63	0.78
	Z2.5	1.65	1.50	1.22	1.14	1.56	1.45	2.06	1.85	1.42
	Z2.6	1.24	1.21	1.15	0.98	1.46	1.15	1.37	1.26	1.17
	Z2.7	1.06	1.28	0.59	1.10	1.04	0.93	1.15	0.99	1.50
**Central**	Z3.1	2.05	1.41	2.42	1.49	1.38	1.52	1.46	2.49	1.66
	Z3.2	1.00	0.83	1.24	0.83	1.12	1.09	1.02	1.07	0.66
	Z3.3	1.94	1.39	4.43	1.20	1.66	1.48	1.15	2.12	1.56
	Z3.4	1.72	1.20	2.02	1.62	1.21	1.80	4.55	1.67	2.60
	Z3.5	1.79	1.04	2.63	1.51	1.40	1.10	1.85	2.12	1.17
	Z3.6	2.04	1.05	2.79	1.71	1.15	1.50	2.66	2.48	1.60
	Z3.7	0.71	0.94	0.79	0.75	0.83	0.81	0.50	0.64	0.60
**Edmonton**	Z4.1	0.66	0.84	0.38	0.84	0.80	0.61	0.46	0.59	0.77
	Z4.2	0.74	0.91	0.43	1.05	0.84	0.71	0.54	0.65	0.90
	Z4.3	0.54	0.80	0.24	0.75	0.73	0.43	0.40	0.45	0.57
	Z4.4	0.46	0.61	0.17	0.55	0.68	0.32	0.31	0.40	0.44
	Z4.5	0.96	0.82	1.30	1.30	1.10	0.68	0.36	0.89	0.95
	Z4.6	0.69	0.69	0.45	0.77	0.86	0.36	0.39	0.73	0.47
	Z4.7	0.96	0.92	1.41	0.75	1.11	0.61	1.45	0.90	1.05
	Z4.8	0.90	0.65	0.77	1.07	1.13	0.70	1.15	0.78	1.43
	Z4.9	0.61	0.79	0.23	0.85	1.04	0.57	0.32	0.44	0.85
**North**	Z5.1	2.06	1.14	4.49	1.69	1.57	1.62	1.71	2.35	1.32
	Z5.2	2.83	1.59	4.59	2.02	1.30	3.30	4.63	3.29	2.86
	Z5.3	2.50	1.06	4.80	2.63	1.53	2.33	2.30	2.94	1.49
	Z5.4	3.11	1.43	4.54	3.58	1.23	3.22	4.57	3.57	3.21
	Z5.5	1.40	0.98	2.67	1.16	1.04	2.04	0.44	1.53	1.10
	Z5.6	1.49	1.04	3.00	1.15	1.13	2.19	0.64	1.63	1.09
	Z5.7	1.34	0.96	1.76	1.88	1.12	1.02	1.20	1.38	1.20

^1^ Heat map gradient goes from lower (green) to higher (red) SPR. oLRTI = Other acute lower respiratory tract infections. oURTI = Other acute upper respiratory tract infections.

**Table 3 ijerph-18-08347-t003:** Spearman correlation index between the SPRs of healthcare utilization events for respiratory illnesses and the material and social deprivation levels of all subzones.

	Material Deprivation	Social Deprivation
SPR total	0.48 *	−0.29
SPR asthma	0.44 *	−0.05
SPR bronchitis	0.36 *	−0.33
SPR bronchiolitis	0.36 *	−0.23
SPR croup	0.40 *	−0.36 *
SPR influenza	0.47 *	−0.16
SPR oLRTI	0.51 *	−0.16
SPR oURTI	0.46 *	−0.31
SPR pneumonia	0.40 *	−0.22

* Indicates correlation coefficients significant at the 0.05 level or lower.

## Data Availability

Data cannot be shared publicly because it is held securely in coded form at Alberta Health Services. Alberta Health Services is the legal custodian of the original data. Alberta Health Services’ policies and acts (e.g., Health Information Act of Alberta) guarantee the security, privacy and confidentiality of the patient data. Data agreement with Alberta Health Services prohibits researchers from making the dataset publicly available. Access to data may be granted to those who meet pre-specified criteria for confidential access. Data are available from Alberta Health Services Provincial Research Data Services for researchers who meet the criteria for access to confidential data. The data underlying the results presented in the study are available from Alberta Health Services’ (AHS) Health System Access (HSA): https://www.albertahealthservices.ca/research/page8579.aspx (accessed on 28 July 2020). More information at: research.administration@ahs.ca.

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
