# Peer review of "Geography as a Determinant of Health: Health Services Utilization of Pediatric Respiratory Illness in a Canadian Province"

_ijerph, 2021, doi:10.3390/ijerph18168347_

Round 1

Reviewer 1 Report

While reading the manuscript another alternative  title comes to mind, for example: Selected determinants of health services utilization of pediatric respiratory illnesses: case study of Alberta. While studying the aforementioned issue, authors have focused on material and social deprivation determinants of health services utilization of pediatric respiratory illnesses in the given area. At the same time, pediatric respiratory illnesses could be determined by broad spectrum of factors, e.g. ecological situation in the macro region and in the place of residence, cultural determinants, access to the primary and specialist healthcare  etc. Adding these determinants into the study  would significantly enrich the study. Moreover  in the discussion part it would be interesting to compare the results of the study with results in other countries with similar socioeconomic situation. 

Author Response

Thank you for this suggestion. We have changed the title of the manuscript to: “Geography as a Determinant of Health: Health Services Utilization of Pediatric Respiratory Illnesses in a Canadian province”

The provincial health databases that supported our research do not systematically collect information on other social determinants that play an important role in creating health inequalities. We have acknowledged this limitation in the Discussion section. Refer to manuscript starting at line 317.

A review of the literature suggests that limited research exists on geography and deprivation for pediatric respiratory illnesses, and we hope that this study will open doors for future analyses to allow for comparison with higher income countries such as Canada. 

Reviewer 2 Report

General remarks

The paper deals with a very interesting and timely issue— the impact of geography as a determinant of health, especially between rural and urban spaces. This is a relative blind spot in the scholarly literature, so the paper contributes to filling this gap. That said, a few things need tightening though to further enhance the paper’s quality.

Introduction

- The paper is comprehensively referenced to existing literature.

-The authors should insert a paragraph after line 69 that states how the paper is organised, as this will guide readers. For instance, ‘the rest of the paper is organised as follows: the next section outlines the material and methods, then we present our results…’ In other words, this should be the last paragraph before you move to the next section (that is, ‘Materials and Methods’).

Conclusions

-The conclusion is too brief and does not do justice to the richness of the paper. The authors should start by restating the purpose of the paper and proceed to highlight a few key findings.

-Also, the authors should briefly discuss the implications of their findings. Specifically, what should the provincial government, healthcare facilities, and other relevant stakeholders do in light of these findings?

Author Response

Thanks to the reviewers for the constructive suggestions to improve the quality of the manuscript.

Thank you for the suggestion about stating how the paper is organized. We are following the journal templates with headings that clearly indicate the sections in which the manuscript will be divided. We believe adding a paragraph about the paper organization is a bit redundant. Changes were not made.

The purpose of the study was restated in the conclusion, and we added a bit more clarity to some key findings that were mentioned in both Discussion and Conclusion sections. Refer to the manuscript starting at line 350. Thank you for this suggestion.

Implications of this study have been added to the Discussion to address the reviewer comment about what should the provincial government, healthcare facilities, and other relevant stakeholders do in light of our findings. Refer to the manuscript from lines 331 to 340